# Structural Lesions of Proteins Connected to Lipid Membrane Damages Caused by Radical Stress: Assessment by Biomimetic Systems and Raman Spectroscopy

**DOI:** 10.3390/biom9120794

**Published:** 2019-11-27

**Authors:** Armida Torreggiani, Anna Tinti, Zuzana Jurasekova, Mercè Capdevila, Michela Saracino, Michele Di Foggia

**Affiliations:** 1Istituto I.S.O.F. (C.N.R.), via P. Gobetti 101, 40129 Bologna, Italy; michela.saracino@isof.cnr.it; 2Dipartimento di Scienze Biomediche e Neuromotorie, Università di Bologna, via Belmeloro 8/2, 40126 Bologna, Italy; anna.tinti@unibo.it (A.T.); michele.difoggia2@unibo.it (M.D.F.); 3Department of Biophysics, Faculty of Science, P.J. Safarik University, Jesenna 5, 04001 Kosice, Slovakia; zuzana.jurasekova@upjs.sk; 4Center for Interdisciplinary Biosciences, Technology and Innovation Park, P. J. Safarik University, Jesenna 5, 04001 Kosice, Slovakia; 5Departament de Quimica, Facultat de ciencies, Universitat Autònoma de Barcelona, 08193 Bellaterra, Catalonia, Spain; merce.capdevila@uab.cat

**Keywords:** free radical stress, lipid domains, protein damage, Raman spectroscopy, desulfurization, protein:lipid interaction, *cis-trans* isomerization, tandem lesions, thiyl radicals

## Abstract

Model systems constituted by proteins and unsaturated lipid vesicles were used to gain more insight into the effects of the propagation of an initial radical damage on protein to the lipid compartment. The latter is based on liposome technology and allows measuring the *trans* unsaturated fatty acid content as a result of free radical stress on proteins. Two kinds of sulfur-containing proteins were chosen to connect their chemical reactivity with membrane lipid transformation, serum albumins and metallothioneins. Biomimetic systems based on radiation chemistry were used to mimic the protein exposure to different kinds of free radical stress and Raman spectroscopy to shed light on protein structural changes caused by the free radical attack. Among the amino acid residues, Cys is one of the most sensitive residues towards the attack of free radicals, thus suggesting that metal-Cys clusters are good interceptors of reactive species in metallothioneins, together with disulfides moieties in serum albumins. Met is another important site of the attack, in particular under reductive conditions. Tyr and Phe are sensitive to radical stress too, leading to electron transfer reactions or radical-induced modifications of their structures. Finally, modifications in protein folding take place depending on reactive species attacking the protein.

## 1. Introduction

The *cis* geometry of double bonds in lipids is a crucial feature for the functionality of cell membrane because it regulates the membrane properties such as permeability and fluidity, and membrane homeostasis is obtained by a precise balance between saturated and *cis*-unsaturated structures. On the contrary, *trans* geometry is unusual in nature and only some bacteria make use of an enzymatic cis-trans isomerization during the adaptation response [1].

The *cis-trans* isomerization can affect lipid assembly. Since the *trans* lipid geometry resembles that of saturated lipids [2], the geometrical isomerization can be considered equivalent to the decrease of the degree of unsaturation in the cell membrane [3]. In addition, the *trans* lipid incorporation in membranes is also linked to a recently growing concern in nutrition [4].

The real culprits for the *cis-trans* isomerization in vivo have not been demonstrated yet, although many results support the correlation between the origin of endogenous formation of *trans* lipids and radical stress. Thiyl radicals RS^•^ are the most likely candidates because of the efficiency of the thiyl radical-catalyzed *cis-trans* isomerization in vitro, and the presence of many sulfur-containing compounds within the cell. The role of thiyl radicals as radical stress inducers has emerged in the last decade [5,6], counteracting the long-standing and most well-known beneficial effects of thiols. The damage effect starts from the consideration that, in the so-called ‘repair reaction’, Cys residues can efficiently stop the radical cascade, by donating a hydrogen atom and trapping the radical intermediate of the chain. However, after this hydrogen donation, the thiol (RSH) is transformed into its corresponding thiyl radical (RS^•^), which can, in turn, attack a substrate.

Possible scenarios for the production of diffusible thiyl radicals were expanded by the discovery of free-radical damage to sulfur-containing proteins resulting in consequent isomerization of membrane lipids. In fact, a radical reaction with a primary target amino acid could lead to secondary radicals. These may engage in radical- and/or electron-transfer processes with additional protein amino acids, so that the ultimate location of a radical damage may evolve potentially far from the site of initial radical attack [7]. Thus, tandem lesions can occur from a single radical event, and at the same time produce reactive species able to damage another molecule. This tandem process can involve two different cell compartments, such as in case of the lipids with DNA molecules [8] or with proteins [9].

Under radical stress, thiyl radicals can be readily generated from a free thiol group by hydrogen abstraction, by cleavage of disulfide linkages and by the attack to methionine (Met) residues and metal-sulfur clusters. Thus, a desulfurization process takes place, coupled with the formation of S-centered radicals (CH_3_S^•^ or S^•–^), which in turn can migrate from the aqueous to the lipid bilayer, causing *cis-trans* isomerization of unsaturated fatty acid residues through their reversible addition to double bonds (Scheme 1a) [10,11,12,13,14,15]. 

The desulfurization processes correspond to a mutation of the natural sequence of proteins. For example, Cys residues can be converted into dehydroalanine residue, or if a disulfide bods is attacked, one cysteine (Cys) involved in the S-S bridge is mutated into alanine (Ala), whereas Met is mutated into α-aminobutyric acid (Aba) (Scheme 1b) [16]. This amino acid is not genetically coded, therefore the conversion of Met into Aba is analogous to a post-translational modification into another natural structure, whose biological consequences are not known.

The most studied intermediates known to cause protein damage are reactive oxygen species (ROS) and in particular OH^•^ radicals [17,18], whereas the reductive stress has been less considered. However, not only oxidative species but also reductive ones, such as hydrogen atoms and solvated electrons, have been found to cause deleterious effects in cells [19,20,21]. On this basis, it is evident that specific research has to be conducted in order to connect the chemical reactivity to the biological environment, including the scenario of reductive radical stress, less considered than the oxidative one.

In order to elucidate some possible scenarios for the production of diffusible thiyl radicals, our research strategy combined different approaches. Biomimetic systems based on radiation chemistry were used to mimic protein exposure to different kinds of free radical stress. In addition, model systems constituted by protein and unsaturated lipid vesicles were used to get information on how the initial radical damage on protein can be eventually propagated to the lipid compartment. Finally, the protein structural changes caused by the free radical attack were analyzed by using Raman spectroscopy, that can shed light on the preferential site of the radical attack and on the radical-induced modifications in protein folding.

Two kinds of sulfur-containing proteins, serum albumins and metallothioneins, were chosen to better connect the chemical reactivity with membrane lipid transformation. Human serum albumin (HSA) is one of the most important plasma antioxidants in protecting key regulatory proteins, has many functions (i.e., ligand-binding and antioxidant) and it is rich in sulfur-containing residues (Table 1). It contains Met residues, disulfide bonds (17 S-S) and one free thiol group (CySH). Its bovine counterpart (BSA) slightly differs from HSA in the sequence (76% is identical), as reported in Table 1 [22]. As regards metallothioneins (MTs), they are low molecular weight, sulfur-rich proteins (*thio* means sulfur), characterized by a high content of Cys (almost a third of the total amino acids) and a high capacity for binding metals by formation of metal-thiolate (metal-Cys) clusters. They are present in all living organisms and have multifunctional roles [23,24], among them scavengers to various radicals and ROS [25,26]. 

In this work we present a study of two homologous plant MTs (from cork oak and soybean) and two serum albumins under oxidative and reductive radical stress in order to compare their capability to scavenge reactive radicals, to transfer eventually a damage to lipid compartment, as well as to evaluate the consequence on the protein structure. In fact, the connection of protein reactivity with membrane transformation can give a contribution to the puzzling context of radical stress occurring to biomolecules.

In native M-MTs complexes (M standing for metal) S-S bonds are almost completely absent because all the thiol groups of Cys residues are deprotonated and bound to metal ions. In addition, M-MTs contain Met residues, and sometimes also labile sulfide anions (S^2−^) as non-proteic ligands [27] that could generate diffusible sulfhydryl radicals and promote isomerization of double bonds in membrane lipids.

The presence of sulfur-containing amino acids, methionine (Met) and cysteine (Cys), as well as sulfide anions (S^2−^) in the same molecular aggregate makes this system a very interesting case to study in the contest of the tandem protein-lipid damages [28]. Two plant MTs from cork oak (*Quercus suber* —QsMT) and soybean (*Glycine max*—GmMT) (the subfamily MT2) were chosen because of their structural homology: both metal-protein complexes present two Cys-rich domains separated by one Cys-devoid domain; the length of the two MTs and the Cys number is about the same (Table 1). In addition, Cys residues are in a similar sequence position and there are similarities also between their metal-clusters [29,30]. 

## 2. Materials and Methods 

### 2.1. Protein Models and Their Analysis

Fatty acid-free human and bovine serum albumin (HSA and BSA) were purchased from Sigma-Aldrich (trought Merck Life Science S.r.l. 20149, Milan, Italy) and used without further purification. HSA or BSA (1 mg) were dissolved in 1mL of tri-distilled water or aqueous 0.2 M t-BuOH solution.

The two homologous plant MTs from cork oak (*Quercus suber*—QsMT) and soybean (*Glycine max* —GmMT) (the subfamily MT2) were recombinantly synthesised in *E.coli*, following a rationale based on the heterologous synthesis of the protein in *E.coli* cells harbouring the cDNA encoding for the desired MT, in its wild-type form, in form of split domains or site-directed mutants. The metal bound to the aggregates depends on which metal is supplemented to the media where the bacteria are growing, so the synthesis of the metallic aggregates takes place inside the cell. The use of this genetic engineering approach is an optimal way to obtain significant amounts of highly pure MTs that does not require any further manipulation, being representative of in vivo folding processes since they are formed into a cell environment. 

By sonication, centrifugation, and chromatographic procedures, the M-MT aggregates are obtained in purity and amounts enough for spectroscopic structural studies. Full details of the synthesis and purification procedures have been extensively described in [31,32]

The S and Zn content of all Zn-MTs preparations was analyzed by inductively-coupled plasma atomic emission spectroscopy (ICP-AES) in a Polyscan 61E (Thermo Jarrell Ash, Franklin, MA, USA) spectrometer, measuring S at 182.040 nm and Zn at 213.856 nm. Molecular mass determinations were performed by electrospray ionization time-of-flight mass spectrometry on a Micro Tof-Q instrument (Bruker, Bremen, Germany) interfaced with a Series 1100 HPLC Agilent pump (Agilent Technologies, Waldbronn, Germany). The content of acid-labile sulfide ligands (S^2−^), referred to the MT concentration measured by acid ICP-AES, was quantified by gas chromatography—flame photometric detection (HP 5890 Series II gas chromatograph coupled to a FPD80 CE Instruments –Thermo Finnigan- detector, Thermo Electron Corporation) after strong acidification (generation of H_2_S). Finally, the molecular mass of the M-MTs species is obtained by electrospray ionization mass spectrometry (MicroToF-Q Instrument – Bruker, Bremen, Germany) [30,31].

The recombinant Zn-MTs used in this work contain the same amount of metal (metal-to-protein ratio is 3.5), whereas the sulfide ions are present only in the isoform from *Quercus suber* (sulfide-to-protein ratio is 1). 

To study the structural properties of polypeptides and proteins many different chemico-physical techniques can be utilized, i.e., nuclear magnetic resonance and X-ray diffraction. The disadvantage of these two techniques is that they require large amounts of sample and are slower than the optical methods. To obtain information on the radical-induced products gas chromatography–mass spectrometry (GC/MS) and HPLC techniques have been generally used, but both methods require isolation and hydrolysis of the samples prior to product quantification. In this context Raman spectroscopy can give a useful contribution since it is capable of providing a rapid and non-destructive analysis of the samples, and it can detect several structural features at the same time. Raman spectroscopy involves the analysis of scattered photons from a laser beam focused into the sample. A small percentage of the scattered photons exchanges energy with the vibrational energy levels (or, crudely, the ‘vibrations’) of the molecules. These vibrations are a function of molecular conformation, of the distribution of electrons in the chemical bonds, and of the molecular environment. Thus, a Raman spectrum provide information on local sites in a much larger macromolecule, and can be of help in identifing the favourite sites of the radical attack and radical-induced modification in protein folding.

One of the greatest advantages of Raman spectroscopy is the lack of sample preparation needed prior to analysis. Because Raman scatter originates from the surface of a sample, there is no concern with sample thickness or size, it requires a low amount of sample.

To obtain samples suitable for the Raman analysis avoiding spectral interference by the Tris buffer used during the M-MTs synthesis, a dialysis-lyophilization protocol was implemented prior to the spectroscopic measurements [30]. Raman spectra were recorded before and after irradiation of the aqueous solutions by a Bruker IFS 66 spectrometer equipped with a FRA-106 Raman module and a cooled Ge-diode detector, in order to follow the changes in the protein structure and the microenvironment of amino acidic residues, resulting from the exposure to radical species. The excitation source was a Nd^3+^-YAG laser (1064 nm), the spectral resolution was 4 cm^−1^ and the total number of scans for each spectrum was 6000. The laser power on the sample was about 100 mW. No carbonization effect was observed on focused part of the samples, as already tested in the previous studies [13,14,15,29,30]. Raman spectra were obtained on lyophilized samples, before and after irradiation of the aqueous solutions in order to improve the signal/noise ratio that was low in aqueous solutions due to the interference from protein fluorescence and to avoid any eventual post-irradiation effect in the protein samples, since in radiolysis of water, also nonradical oxidants (H_2_O_2_), together with HO•, eaq- and H• short-lived species, are produced. Lyophilization was performed on a Modulo 4K Freeze Dryer equipped with a RV8 Rotary Vane Pump (Edwards, Fisher Scientific, Pittsburgh, PA, USA). The lyophilized products were kept at −80 °C until use. 

The curve fitting analysis for the secondary structure determination was accomplished by using the OPUS/IR v5.0 program (Bruker Italia, Milan, Italy), using the Levenberg–Marquardt algorithm. Peak component number and position were identified by fourth-derivative spectra, smoothed with a 13-point Savitsky–Golay function. These band positions were used as the initial guess for the curve fitting of the original spectra. Best curve fitting was obtained at the lowest possible χ^2^ values. The Raman component profiles used in the curve fitting was a linear combination of Lorentzian and Gaussian functions. The content of secondary structure elements was calculated from the integrated intensities of the individually assigned bands and their fraction of the total intensity. All the components were assigned to a particular secondary structure on the basis of the literature [33,34,35,36,37]. 

### 2.2. Radiolytic Production of Transients

Reactive species generation, mimicking the conditions of an endogenous radical stress, was obtained by using γ-radiolysis of aqueous solutions (^60^Co-Gammacell), which allows selecting the reacting species by changing the appropriate experimental conditions. 

Exposure to irradiation conditions, such as radiolysis, has been largely used for the study of oxidative stress, thus providing a very useful tool for investigating mechanistic and structural features of the damages, i.e., to amino acids [17,18]. 

When diluted aqueous solutions are irradiated, practically all the energy absorbed is deposited in water molecules giving rise to water radiolysis. Thus, the chemical changes in solutes are brought about indirectly by the three short-lived species produced, hydrated electron (e_aq_^−^), hydroxyl radicals (^•^OH), and hydrogen atoms (H^•^), together with H^+^ and H_2_O_2_, (Equation (1)) [38]. The first three reactive species are expected to react very fast with peptides and proteins. For example, the rate constants for e_aq_^−^ and ^•^OH species with serum albumin are 8.2 × 10^9^ and 7.8 × 10^10^ M^−1^ s^−1^, respectively [11,38]. 

Direct action due to energy deposited directly in the solute is not important when the solute concentration is below 100 mol/m^3^ [39].
(1)$$H_{2}O\;∼∼∼\overset{\gamma }{\to }{\;e}_{\mathrm{aq}}^{-}\;\left(0.27\right),{\;\mathrm{HO}}^{•}\;\left(0.28\right),{\;H}^{•}\;\left(0.06\right),{\;H}^{+}\;\left(0.27\right),{\;H}_{2}O_{2}\;\left(0.07\right)$$

The values in parentheses represent the radiation chemical yields expressed in terms of G-value (µmol·J^−1^), corresponding to the yield of radicals per 10 eV of absorbed energy. By saturating with N_2_O, e_aq_^−^ are efficiently converted into the oxidizing HO^•^ radical (Equation (2)) and mainly oxidative stress conditions are obtained.
(2)$$e_{\mathrm{aq}}^{-}+N_{2}O+H_{2}O\;\to {\mathrm{HO}}^{•}+N_{2}+{\mathrm{HO}}^{-}$$

Under these experimental conditions, used in model studies of oxidative damages occurring in vivo [40,41], HO^•^ and H^•^ radicals account for 90% and 10%, respectively, of the reactive species. 

It is possible to investigate the reactivity of only reductive species (e_aq_^−^ and H^•^) and their consequences on protein structure by adding an alcohol, such as *t*-BuOH (0.2 M), to the protein solution, and irradiating in an oxygen-free atmosphere. Under these experimental conditions, HO^•^ radicals are efficiently scavenged, and e_aq_^−^ and H^•^ account for about 80% and 20% of the reactive species, respectively. Finally, by using N_2_O-saturated solutions in combination with *t*-BuOH as additive, the reactivity of only H^•^ atoms can be investigatedsince the presence of *t*-BuOH limits the eventual observations of undesired polypeptide modifications resulting from HO^•^ radicals. 

### 2.3. Biomimetic Models

Biomimetic models can be very useful to address lipid-protein damage, in order to test the capability of proteins under free radical stress to damage the lipid cell compartment and to extrapolate the mechanistic information to the more complex situations present in vivo. Monolamellar vesicles, formed by the extrusion methodology and made of phospholipid containing an unsaturated fatty acid moiety in the *cis* configuration (such as oleic acid), are simple but appropriate models of natural membranes. They are a good biomimetic model for the double bond isomerization process if used as aqueous suspensions in the presence of peptide/proteins and the reactive radical species since they are mimicking two biological compartments, the membrane and the extracellular protein domain. In detail, we used proteins at micromolar concentration levels added to a millimolar vesicle suspension; the vesicles being formed by 1-palmitoyl 2-oleoyl phosphatidylcholine (POPC), where one of the two fatty acid chains presents the double bond in the *cis* configuration. Aliquots of the suspensions are saturated with an appropriate gas (N_2_O or Ar) prior to γ-irradiation at different doses, in order to selectively form the reactive radical species. After trans-esterification of the phospholipids to the corresponding fatty acid methyl esters [42], the *trans* isomer content, calculated as oleate/elaidate (*cis*/*trans*) ratio, is obtained by gas chromatographic (GC) analysis [43]. Based on the palmitic moiety of POPC, which served as the internal standard, trans isomers in our experiments were formed quantitatively and no side-reactions occurred.

GC is the easiest and fastest technique for fatty acid analysis. GC analysis requires the fatty acid-containing lipids to be converted to more volatile compounds, such as methyl ester or other ester derivatives (FAMEs). For separation of all of FAMEs components in a mixture, the choice of the chromatographic column and the carrier gas play a crucial role. In this study, the GC analysis were performed by the Agilent 6850 (Milan, Italy) GC system equipped with a 60 m × 0.25 mm × 0.25 μm (50%-cyanopropyl)-methylpolysiloxane column (AgilentJ&W DB-23, Cernusco sul Naviglio MI, Italy) and a flame ionization detector. A constant pressure mode (29 psi) was chosen using helium as the carrier gas [44,45]. 

As regards the experiments with biomimetic models, 30 µM Zn^II^-MTs or 9.2 μM HSA/BSA aqueous solutions were added to 2 mM liposome suspensions of POPC containing large unilamellar vesicles (diameter = 100 nm) and γ-irradiated. 100 µL aliquots of the protein–lipid suspension were withdrawn at different irradiation times for lipid isolation, following the procedure described elsewhere [46]. When necessary, *t*-BuOH was also added to reach a final concentration of 0.2 M, a value known to be compatible with vesicle stability. Control experiments in the absence of proteins confirmed that *trans*-isomer formation is less than 0.2% after exposure to 500 Gy. All set of experiments were replicated twice (M-MTs) or three times (HSA/BSA), depending on the availability of sample amount for the anaysis. The error in the determination of the *trans* isomer percentage ranged from 2% to 6% (the lowest errors in the case of serum albumins).

It is worth underlining that even very low levels of protein modifications produced by reactive species can be detected by the liposome vesicle reactivity. This is due to an amplification effect given by the catalytic cycle of the thiyl radical-based *cis-trans* isomerization [47]. 

## 3. Results and Discussion

### 3.1. Damage Transfer from Protein to Lipid 

In order to obtain information on protein modifications caused by the HO^•^ and H^•^ attack and verify the eventual release of sulfur-centered radicals from the MTs isoforms and serum albumins, lipid-protein systems were used. All the four examined proteins under the attack of mainly HO^•^ radicals species were able to cause the *trans*-isomerization of the naturally occurring fatty acids in a dose-dependent manner (Figure 1a,b) [48,49,50,51]. These results confirm once again that under radical stress thiyl radicals can be readily generated from the cleavage of disulfide linkages (the latter are abundant in serum albumins and almost absent in MTs), the attack to Met residues or metal-thiolate clusters, where the sulfur atom of one Cys can bind to one or two metals.

The comparison between proteins with a high percentage of homology shows the importance of the sequence for the reactivity and the potential to generate small radicals involved in tandem damage. In fact, both Zn^II^-GmMT2 and HSA, containing a higher number of sulfur-containing residues than Zn^II^-QsMT2 and BSA, respectively, have a higher capacity to induce *cis-trans* isomerization than the other two proteins. This is particularly evident at the highest dose when serum albumins and the metal complexes have almost lost their native folding, making the sulfur-containing moieties more accessible to the free radical attack and the lipid damage strictly connected to the amino acid content in the sequence. In Zn^II^-MTs systems the sulfur-centered radicals can be generated by the attack of HO^•^ to Cys bound to the metal ions and H^•^ to Met residues, respectively (Scheme 2).

As regards BSA and HSA, thiyl radicals under anaerobic conditions are mainly produced by the attack of HO^•^ and H^•^ to disulfide bonds, as shown in Scheme 3, but they can be also generated from the fragmentation of Met residues caused by the H^•^ attack (Scheme 2). 

All the four protein models showed to have a high capacity to induce *cis-trans* isomerization under reductive radical stress conditions (Figure 1c,d). For example, the formation of *trans* isomer caused by the attack of mainly HO^•^ radicals towards Zn^II^-GmMT2 is about 20%, whereas it increases up to about 50% when e_aq_^−^ and H^•^ are the reactive species. This result suggests that reducing species are very specific damaging agents for sulfur-containing residues, therefore contributing significantly to the transfer of the protein damage towards the lipid domain.

This can be explained by considering that the attack of HO^•^ radicals can occur at the level of sulfur-containing residues, aromatic amino acids, aliphatic side chains, and, consequently, multiple products can be generated. For example, the attack towards sulfur-containing residues can produce oxidation derivatives such as disulfide bridges and sulphoxides, whereas the withdraw of an H atom from the α-carbon can give rise to the release of thiyl radicals capable to diffuse into lipid bilayers and induce secondary damage.

Under reductive stress conditions the higher isomerization yields obtained from Zn^II^-GmMT2 and HSA in comparison with Zn^II^-QsMT2 and BSA can be attributed to several factors, such as the higher number of Met residues in Zn^II^-GmMT2 and HSA and a slight different folding of the corresponding protein systems which can play a role in the accessibility to free radicals to reach sulfur-containing residues. In addition, the presence of two Tyr residues in Zn^II^-GmMT2, absent in Zn^II^-QsMT2, can contribute to the higher capability of Zn^II^-GmMT2 to transfer the damage to lipid domains. In fact, although the benzene ring of both Tyr and Phe is known to react with H^•^ atoms, leading to the formation of H-ring adducts, only Tyr has been reported to be involved in side-chain reactions where Tyr radicals can be repaired by Cys, yielding thiyl radicals [17]. 

In order to demonstrate which reducing species is more active regarding isomerization yield, experiments only in the presence of H^•^ radicals have been performed with BSA and HSA. It clearly appeared that this species is able to promote the isomerization in an extremely efficient way (i.e., at the highest dose the percentage of *trans* isomer is over 75% for both proteins) (Figure 2). Thus, as regards serum albumins, there is not a synergic contribution of hydrogen- and electron-adduct on the protein system to the formation of diffusible species responsible for the *cis-trans* isomerization. On the contrary, this synergy was found for the M-MTs systems [48].

### 3.2. Protein Analysis by Raman Spectroscopy

The changes in the protein structure and the microenvironment of amino acidic residues, resulting from the exposure to radicals, can be followed by Raman spectroscopy. Indeed, this vibrational technique has become a versatile tool in protein science and its application in characterizing protein in biotechnology, pharmaceutical production, and food industry has been developed [52]. In particular, it can be used to quantify the contribution of distinct secondary structure motif to the overall protein structure. The best-studied Raman band of proteins is the amide I band appearing between 1630 cm^−1^ and 1700 cm^−1^, which arises from the stretching vibration of the peptide C=O group.

#### 3.2.1. Protein Secondary Structure Change

In order to evaluate the damage performed by HO^•^ and H^•^ radicals on the protein structure, Raman spectra were recorded after exposure at different doses under N_2_O-saturated conditions (i.e., Zn^II^-GmMT2 at 100 and 300 Gy). The shape of Amide I band in the Zn^II^-GmMT2 spectrum significantly changed with the appearance of shoulders at higher and lower wavenumbers respect to the band maximum (Figure 3), indicating differences in the secondary structure contents after the radical stress exposure. In particular, the new Amide I component, giving rise to weak shoulder at about 1686 cm^−1^, is indicative of an increase in the beta-turns content [34]. 

Analogously, weak modifications in the Amide I spectral profile were observed in HSA undergone to 300 Gy irradiation dose (Figure 4); under reductive radical stress conditions, the appearance of a shoulder at about 1680 cm^−1^ as well as the increase of the Amide III component at about 1280 cm^−1^ are indicative of an increase in the beta-turns content (Figure 4). 

To obtain a semi-quantitative evaluation of the conformational changes upon irradiation, a curve fitting analysis can be performed. The neat Zn^II^-GmMT2 system was found to contain mainly β-strands and β-turns [30], as well as a significant contribution from β-strands was found also in Zn^II^- and Cd^II^-QsMT2 complexes [53]. Both mainly oxidative and reductive radical stress conditions cause a loss in β-strands and a relevant gain of β-turns in these protein systems (Table 2). A more extreme unfolding, with a notable increase also of random coil structure, takes place in Zn^II^-GmMT2 under strictly reductive radical stress conditions (e_aq_^−^ and H^•^—Table 2), similarly to the behavior of the Zn^II^-QsMT system, where the exposure to reductive species induces an increase in the percentages of β-turn conformation, according to the theoretical prediction reported for the normal modes of β-turns [53,54]. In the case of QsMT, the increase of random coil conformation in reducing conditions was observed for Cd^II^-QsMT [53]. 

As regards serum albumins, the curve fitting analysis of Raman spectra showed that both mainly oxidative and reductive radical stress conditions cause a loss in α-helix and an increase in the β-sheets content (Table 3). 

Also in this case a more significant unfolding of the helical structure takes place under the attack of hydrogen atom than of hydroxyl radicals. This is in agreement with the literature where H^•^ atoms have been found to have a high ability in denaturing protein structure [55,56].

The conformation changes induced by the reductive stress conditions can be due both to the addition of the solvated electron to the backbone carbonyl groups, that can eventually result in the cleavage of peptidic bonds [17], and to the selective attack of H^•^ atoms at different sites. 

#### 3.2.2. Specific Amino Acid Residues Damage

As regards the side chains, aromatic residues, in particular Tyr, have resulted to be among the most sensitive residues towards radical attack. This conclusion was drawn out from the analysis of the doublet at 850–830 cm^−1^ that depends on the state of the hydrogen bonding involving the OH group of Tyr and from other Tyr bands, such as that at 645 cm^−1^ [33]. For example, GmMT2 contains two Tyr and one Phe residues that give rise to many Raman bands due to aromatic ring vibrations (Figure 3). Exposure of Zn^II^-GmMT2 to radical stress causes significant changes in the I_852_/I_830_ ratio, indicating that the environment around the Tyr residues has become more hydrophilic and/or Tyr are less bound to COO^-^ of aspartic and glutamic acid residues. In addition, the intensity ratio between the ~645 and ~620 cm^−1^ bands, due respectively to Tyr and Phe residues, significantly decreased by increasing the free radicals concentration, showing a quite good linear correlation (*R^2^* = 0.98) with the radiation dose used (Figure 5). This change could be due to a decrease in the Tyr number that, forming phenoxyl radicals by reacting with HO^•^ radicals, give rise to bi-tyrosine cross-linked derivatives [57].

A similar behavior was shown also by HSA when exposed to the e_aq_^−^ and H^•^ attack (the intensity of the 645 cm^−1^ band strongly decreased, see Figure 4) [36]. The I_645_/I_620_ ratio has been connected to the Tyr/Phe content ratio in some proteins [58], so this change could be indicative of a radical-induced modification of some Try residues into Phe residues. Higher Phe/Tyr ratio has been found in blood plasma of patients with HIV infection, with cancer and trauma, and it has been associated with radical stress due to inflammation and immune activation [59], although the mechanism of the disturbed Phe metabolism has not still completely understood.

Both oxidative and reductive radical stress causes significant changes in the spectral features of the S-S stretching bands of HSA (17 disulfide bridges), even at the lowest irradiation dose (Figure 4), since disulfide bonds rapidly react with HO^•^ radicals, H^•^ atoms and solvated electrons, leading under anaerobic conditions to thiyl radicals RS^•^ (Scheme 2). The latter can regenerate disulfide bridges, that can have different conformation with respect to the initial one (that means different Raman bands) or give rise to lipid *cis-trans* isomerization by producing HS^•^/S^•–^ radicals [36]. These modifications were also confirmed in the case of HSA, by mass spectrometric analysis which evidenced the chemical transformation of one Cys involved in S-S bond into Ala as well as a rapid disulfide scrambling events towards more stable disulfide/thiol population [50]

Moreover, as regards serum albumins, the spectral changes in the Phe bands can be connected with the relevant modifications occurring in disulfide bridges under reductive radical stress conditions. It is known that solvated electron reacts with Phe generating a cyclohexadienyl radical that could be repaired by one-electron oxidation to an acceptor molecule (Scheme 4). Disulfide linkages can be these acceptor moieties since they are known to act as a major sink for electrons arising from electron transfer by reducing species:

Thus, an initial addition of solvated electrons to Phe residues can result in the ultimate reduction of cystine groups, which can then lead to the fragmentation of the disulfide bonds and generation of RS^•^ radicals (Scheme 3).

The occurrence of an intra-protein electron transfer can contribute to obtaining a higher yield of isomerization for HSA, which contains more Phe residues than BSA (Figure 1d and Table 1).

Also in M-MTs systems Cys is among the most sensitive residue towards radical attack, since the radical stress exposure causes a partial rearrangement and deconstruction of metal-thiolate clusters, as shown by the spectral modifications of some Zn-S stretching bands visible at low wavenumbers, typical of M-S stretching (280–350 cm^−1^) [53]. Demetallation reactions of the Zn^II^-MTs systems can also take place, as confirmed by various spectrometric techniques such as CD and ESI-MS [48,51]. 

As regards the Met residues, they are sensitive to the free radical attack in all conditions, as indicated for example by the decrease in intensity of the 726 cm^−1^ band due to the νC-S bond of Met residues in the Zn^II^-GmMT2 system (Figure 3).

Thus, this investigation underlines the damages that oxidative and reductive radical attacks can cause to the secondary and tertiary structure of proteins and to specific amino acid residues.

## 4. Conclusions

In the context of free radical processes with biological interest, tandem protein–lipid radical damage has been the subject of recent investigations that pointed out its harmfulness in the general scenario of establishing the consequences of radical stress. Lipid vesicle suspensions containing a protein have been found very useful in the initial identification of the coupled lipid–protein damage since the amplified membrane damage is the final effect of these reactive pathways.

The results of γ-irradiation experiments reported in this paper can give a contribution to the knowledge of free radical-induced modifications in the structure of sulfur-containing proteins, resulting in consequent isomerization of membrane lipids. 

Radical damages are not randomly distributed along the polypeptide chain, but specific damages occur at sensitive amino acid sites. Met residues are one of the main sites of the attack and play an important role in the tandem albumin-lipid damage under reductive conditions by producing sulfur-centered radicals by means of a desulfurization reaction. In the case of serum albumins, this role becomes more relevant than that of disulfide bridges, which act as efficient trapping centers of reducing species, by direct trapping of electron transfer within the protein. Thus, an initial addition of solvated electrons to aromatic side chains of Tyr and Phe or to the backbone carbonyl groups of peptide bonds can result in the fragmentation of cystine groups, which can lead to HS^•^/S^•–^ radicals able to promote lipid *cis-trans* isomerization in POPC suspensions or to reform disulfide bonds. 

In M-MTs systems, metal-clusters are good interceptors of free radicals. They react with free radical species, without a complete loss of metals, thus showing their capability to play a protective role against these stressors.

As regards aromatic residues, aromatic side chains of Tyr and Phe have highlighted their sensitivity to radical stress, leading to electron transfer reactions or radical-induced modifications of their structures. Moreover, modifications in protein folding, depending on which reactive species attack the protein, are observed, although both oxidative and reductive radical stress are able to cause a significant loss in α-helix content in serum albumins and an increase in β-turns in M-MTs. 

Since some human pathologies have been associated with damage to unsaturated lipids [60,61], also in this case protein modifications should be used as marker events to eventually monitor pathological conditions. Thus, the chemical evidence acquired so far suggest to combine studies on oxidative and reductive processes to obtain a complete scenario of the reactivity of multifunctional substrates, like proteins, and the free radical stress consequences associated with cellular degeneration, related with aging and pathologies.

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
