# Peer review of "Structural Lesions of Proteins Connected to Lipid Membrane Damages Caused by Radical Stress: Assessment by Biomimetic Systems and Raman Spectroscopy"

_biomolecules, 2019, doi:10.3390/biom9120794_

Round 1
Reviewer 1 Report
In this manuscript the authors address the point of protein and lipid modifications and protein-lipid interactions in response to free radical processes associated with oxidative and reductive stresses. Two type of proteins are used as models, namely serum albumins and metallothioneins. Modifications in their folding characteristics and the impact on the isomerization of membrane lipid chains are studied. The sensitivity of cysteins, methionines and aromatic amino acids is corroborated, along with changes in the frequency of alpha-helix and beta-turn motifs.
Comments :
Scheme 1 is informative and illustrates most part of the introduction. I suggest expanding this Scheme to other processes explained in the introduction, for example the desulfurization of cysteine and methionine and formation of S-centered radicals, with formation of alpha-aminobutyric acid. Alternatively Scheme 2 should be cited in the introduction. A brief description of Raman spectroscopy and gas chromatography procedures including technical information should be included in the materials and Methods section. The results of the rans-isomerization studies shown on Figure 1 should include information about the number of determinations, error bars and eventually a statistical test. The modifications highlighted in the text corresponding to changes in Raman spectra, should be also clearly highlighted on Figure 3 and Figure 4 , for example by means of arrows (shoulders, aromatic ring vibrations, etc. Line 330: The cited Table 1 should be Table 2. The scheme shown between lines 388 and 389 should be numbered and contain a legend.Author Response
Please see the attachment

Reviewer 2 Report
The submitted manuscript reports the study to gain knowledge on the initial damage caused by radicals on proteins and unsaturated lipid vesicles using Raman spectroscopy. The presented results supports the claims made in the manuscripts, but may need additional experiments. I recommend the publication of this manuscript after addression my comments below.
It was stated in the manuscript that the laser power is maintained at 100 mW, which seems to be very high. Could authors comment on how to differentiate the damage caused by radical generated during irradiation and laser power during Raman measurement. I recommend perform these experiments at various laser power and observed the resultant peaks stays same. Does the radical damage is time-dependent? How authors confirmed that the radical are being generated during irradiation experiments?
Round 2
Reviewer 2 Report
I thank the authors for clarifying my comments. The revised manuscript completely addressed my concerns and I suggest this manuscript for publication.